# Correlation between pose estimation features regarding movements towards the midline in early infancy

**Nisasri Sermpon**[1,2], **Hirotaka Gima**[1]*

1 Department of Physical Therapy, Graduate School of Human Health Sciences, Tokyo Metropolitan University, Arakawa, Tokyo, Japan, 2 Faculty of Physical Therapy, Mahidol University, Salaya, Nakhon Pathom, Thailand

* gima@tmu.ac.jp

**Data Availability Statement:** All data files are available from the figshare database (https://doi.org/10.6084/m9.figshare.24143679.v1). All relevant data are within the manuscript and Supporting Information files.

## Abstract

In infants, spontaneous movement towards the midline (MTM) indicates the initiation of anti-gravity ability development. Markerless 2D pose estimation is a cost-effective, time-efficient, and quantifiable alternative to movement assessment. We aimed to establish correlations between pose estimation features and MTM in early-age infants. Ninety-four infant videos were analysed to calculate the percentage and rate of MTM occurrence. 2D Pose estimation processed the videos and determined the distances and areas using wrist and ankle landmark coordinates. We collected data using video recordings from 20 infants aged 8–16 weeks post-term age. Correlations between MTM observations and distance values were evaluated. Differences in areas between groups of videos showing MTM and no MTM in the total, lower-limb, and upper-limb categories were examined. MTM observations revealed common occurrences of hand-to-trunk and foot-to-foot movements. Weak correlations were noted between limb distances to the midbody imaginary line and MTM occurrence values. Lower MTM showed significant differences in the lower part (p = 0.003) and whole area (p = 0.001). Video recording by parents or guardians could extract features using 2D pose estimation, assisting in the early identification of MTM in infants. Further research is required to assess a larger sample size with the diversity of MTM motor behaviour, and later developmental skills, and collect data from at-risk infants.

## Introduction

Infant posture and spontaneous movements hold significance in early development [1]. General movements (GMs) [2], that is whole-body spontaneous movements, have been studied and a method has been developed for detecting neuromotor impairment in the first 5 months post-term age (PTA). The Prechtl general movements assessment (GMA) is an observational method clinically used worldwide. This tool was approved as a validated indicator [3, 4], particularly regarding fidgety movements (FMs, small-moderate speed movements, and varying accelerations in all directions across the neck, trunk, and limbs), for identifying the occurrence of neurodevelopment problems (e.g. cerebral palsy [CP]) or movement disorders [5–8].

**Funding:** This research was funded by the Tokyo Human Resource Fund for the City Diplomacy Scholar and the MEXT Grant-in-Aid for Scientific Research (no.21K11502 to H.G.). The funders had no role in study design, data collection and analysis, decision to publish, or preparation of the manuscript.

**Competing interests:** The authors have declared that no competing interests exist.

Examining additional postural and movement factors could contribute to supporting and reinforcing the early diagnosis of CP [9]. The movement optimality scale [2, 10] was established and revised based on the GMA, with scores ranging from 5 to 28. A decrease in the score may indicate a higher risk of motor or cognitive impairment [11, 12]. Although this assessment result is predominantly based on the presence of FMs, it also considers various concurrent postural and movement patterns observed alongside GMs, which may affect the overall score. These encompass the alignment of the head, body, and limbs, as well as asymmetric tonic neck posture (ATNP) and limb movements, such as movement toward the midline (MTM) (foot-to-foot [FF], fiddling, hand-to-hand [HH], and hand-to-mouth [HM] contact) [2, 12, 13]. A study evaluating concurrent motor and postural patterns with GMs for the early diagnosis of CP found that MTM in both the upper and lower limbs could be considered at fidgety age, following monotonous and stereotyped limb movements [9].

MTM is characterised by limb activities moving against gravity and close to the imaginary midline which symmetrically divides the body into left and right sides. In early infancy, this motor pattern was considered initially developmental toward intentional movement behaviour and oriented toward specific objectives. Lucaccioni et al. [14] expanded the understanding of the typical and age-adequate motor repertoire regarding MTM of the four limbs by observing and identifying the developmental progression in healthy, full-term infants from birth until 22 to 23 weeks PTA. Infants were observed to have 10 MTM in writhing and fidgety movement periods, including: FF, foot-to-leg (FL), HH, hand-to-face (HF), HM, hand-to-trunk (HT), hand-to-leg (HL), manipulation, pedipulation, and fiddling. Progressive increases in frequencies were notably observed, particularly during the age of FMs. The occurrence of MTM in early infancy may indicate movement abilities before reaching goal-directed movements in the future. The frequency of MTM occurrences may be associated with sensory-motor exposure. A lower frequency may signify diminished motor-sensory input, while a higher frequency could imply limited movement patterns due to increased repetition and monotony [15].

Movement analysis exhibits a wide range of diversity, and its level of accessibility depends on numerous elements, encompassing the assessor, the client, available resources, locations, technological advancements, budgetary considerations, and other relevant factors. Previous studies [16–18] demonstrated various movement assessment methods, including visual observation, wearable sensor technology systems, and markerless systems for infant movement monitoring. Using specialized techniques, like three-dimensional (3D) motion capture with wearable sensors, to gather quantitative data can lead to high production costs, necessitating a specific setting and expertise [19]. For example, Kanemaru et al. [20] employed a motion analysis system to examine endpoint movement, focusing on individual limbs and interlimb coordination in healthy infants aged between 2 and 4 months. They analysed these indices as precursors to goal-directed movement. Miyagishima et al. [21] performed a comparison between preterm (corrected age: 1–3 months) and term infants, utilising sensors and a 3D motion capture system, to analyse the characteristics of antigravity spontaneous movements, which encompassed measurements of distance and height for both hands and feet. Marchi et al. [22] examined the predictive capability of quantitative computerised kinematic indices in a group of 12 low-risk term infants using a motion analysis system with six markers. The analysis was done by interpreting indexes of coordination, distance, and global movement quality based on the infants' motions toward the midline.

Recently, human pose estimation has gained widespread use and found applications in various fields, such as sports, home care, work monitoring, and surveillance [23, 24]. The advantages of two-dimensional (2D) and 3D human pose estimation depend on the application objective. 2D human pose estimation is an alternative computer vision technique used to estimate the pose of a person in a 2D image or video [25, 26]. This is a real-time detection method

based on the x and y coordinates of human body joints from RGB (red, green, and blue) images [27]. The study of marker-free infant movement has shown advantages in which infants perform actions naturally with minimal cost, less time consumption, and measurable quantitative data [28, 29]. 3D pose estimation can be achieved using a depth camera (e.g. Microsoft Kinect, Intel RealSense, etc.) or Motion capture system [24, 30]. Similar to marker-based systems, employing markerless 3D techniques is more challenging to implement compared with 2D methods. 3D pose estimation may require proper training, expertise, and high computational complexity for accurate analysis.

There are few MTM studies with both quality and quantity. In the present study, we observed and applied an open source, pose estimation model (google MediaPipe) to quantify MTM in videos of infants taken on their parents' mobile phones at home. MediaPipe pose is a machine-learning solution capable of tracking key body parts in images or videos, and generating 33 pose landmarks presented in either normalized image coordinates or world coordinates [31]. In this study, we simply employed 2D pose estimation and computed distance and area using distal limb landmark coordinates. When infants show MTM in videos, an observable reduction in the distance between the distal part of limbs and the midbody imaginary line is seen. Furthermore, when measuring the spatial area defined by the endpoint of the extremity and midpoint of the body, MTM movements may reduce the enclosed area when compared with instances where the limbs are positioned farther from the midpoint. This study aimed to identify the correlation between features from pose estimation data (i.e. distance and area), and infant movement observations regarding MTM, in infants at an early age (8–16 weeks PTA). We explored the observed MTM values, and the distances calculated through open source 2D pose estimation coordinates, to determine whether a relationship exists or not. Moreover, we identified a disparity in the area between the videos that displayed MTM and those that did not. The hypotheses were that the occurrence percentage and rate per minute of MTM would be associated with the distance between distal joints and the midbody imaginary line. Moreover, the area values in the videos showing lower-part MTM were lower than those in videos without showing MTM, as well as in videos showing upper MTM. This study may be practical and useful to family-centre services for following up infant development with straightforward movement patterns. Utilizing 2D pose estimation can offer a more lightweight and adaptable technology that can be easily applied in everyday practice, especially in low-resource settings, using contemporary technology. The accessibility of such approaches promises to improve access to care for infants with suspected or unidentified risks for neuromotor developmental delay.

## Materials and methods

### Participants and video collection procedure

This study collected video recordings of term infants aged from 8 to 16 weeks PTA without any diagnosed diseases or abnormalities. Participant recruitment was performed using two methods: direct contact with parents or guardians taking care of young infants and asking if they wished to participate in this research voluntarily via email or in person, or via poster invitations in parent support groups on social media asking for voluntary participation. At the onset, the study details were conveyed individually to each parent or guardian. Comprehensive information sheets were provided, and they were actively encouraged to seek clarification. Upon gaining a thorough understanding of the process, they proceeded to electronically complete a consent form, specifying the participant's name, their relationship with the infant, the signatures of the parents or guardians, and the date. Then, we asked parents or guardians to send a video of their infants through their mobile phones using an online private cloud storage

that was secure. This study was approved by the relevant human research ethics committee (approval number: 21092). The recruitment period was from 24th March 2022 to 28th February 2023.

Parents and guardians were instructed to record videos of the infants using the video function of their smartphones, setting the camera to a common configuration of 30 frames per s. First, they were asked to dress infants in a nappy or a bodysuit and place them on a firm mat in the supine position. Next, they had to hold the mobile camera and position it approximately 1 m above the infant, ensuring a top view covering all the infant's limbs. They were instructed to record a video for 2–3 min and avoid interacting with the infant while holding the mobile phone. No specific instructions were given to the guardians regarding when to start recording. Afterwards, the video file was securely transferred to the researcher through cloud storage using multiple security settings on Dropbox. Participants' autonomy and privacy were respected, with careful measures implemented to safeguard the confidentiality of infant data. Following the video collection phase, each video's duration was cut into 2-min clips of active infants lying supine, excluding crying or sleeping during the video. All videos were reformatted with a frame width and height of 1920 × 1080. Consequently, each video commenced by pressing the recording button. The initial 5 s of the video were cut off and subsequently maintained for 2 mins thereafter. Videos were excluded due to visibility issues, such as infants energetically moving their limbs beyond the camera frame, leading to the program's inability to detect corresponding landmark coordinates. Furthermore, if the program failed to execute the code in its entirety, resulting in an incomplete analysis, the video was excluded.

## Procedure of MTM occurrence observation

All included videos were manually observed and scored as absent or present when infants showed MTM during the 2-min video. These were defined as follows [14]: FF elevating both legs, the feet are occasionally brought together with the soles touching each other for a second; FL showing both legs are elevated and the sole of one foot touches the opposite leg for a second; HH showing both hands are brought together and contact each other in the midline for a second; HF showing one or both hands reach and contact the face in the midline for a second; HM showing one or both hands reach the lip with the head aligned at the midline, while the fingers may or may not enter the mouth for a second; HT showing one or both hands reach and touch any part of the trunk for a second; HL showing a hand reach and touch the leg or knee on the same side for a second; FF pedipulation showing feet sliding or grabbing whilst ensuring that the feet remain in contact at the midline and the legs are elevated for a second; HH manipulation showing both hands lift antigravity and brought together at the midline with sliding and grabbing movements for a second; and fiddling, showing one or both hands repeatedly touch, stroke, or grasp an object, often own clothing, for a second. The description of each MTM was clear and easy to follow, thus simplifying the observation process. In addition, the risk of erroneously counting the appearance of an MTM when it was absent was minimised. All videos were examined for frame count, with a total of 3600 frames. The 10 MTM items were observed, with each item requiring a presence of 30 frames continuously to be counted as one second. When an MTM item fell below 30 frames, it was excluded from the count. The occurrence times of each MTM were summed up, and the total occurrence time for the 10 MTMs was calculated. The MTM occurrence percentage for each video was determined using the following equation, where 120 represents the video duration in seconds:

$$\text{MTM occurrence percentage} = \frac{\textit{Total MTM occurrence time (s)}}{120\ (s)} \times 100 \qquad (1)$$

MTM involves contact between limbs or limbs and the body, suggesting a potential connection to sensory-motor input. We aimed to assess the frequency of these occurrences and understand the patterns exhibited with rate per min. The MTM occurrence rate per min was calculated and divided into three groups: total MTM occurrence rate per min, lower-limb MTM occurrence rate per min (FF, FL, and pedipulation), and upper-limb MTM occurrence rate per min (HH, HF, HM, HT, HL, manipulation, and fiddling) [15].

$$MTM \text{ occurrence rate per minute}$$

$$= \frac{MTM \text{ occurrence time } (s) : Total, Lower, Upper \text{ } MTM}{120 \text{ } (s)} \times 60 \tag{2}$$

## Procedure of pose estimation and distance calculation

In this study, Python (version 3.9.12) was applied and we processed 2D pose estimation using the internal MediaPipe function [31] to extract 33 body landmark coordinates (x, y) from each 2-min video. We set the minimum confidence score as 0.5 for pose detection. The landmarks were computed, utilising normalised screen coordinates obtained from the x and y coordinates. Consequently, these points yielded indices with dimensionless values. After running the code, we obtained 118, 800 coordinates in comma-separated value (csv) files. We then checked each file, eliminating those with incomplete data, and excluding the corresponding videos. The selection of wrist and ankle coordinates (landmarks 15 and 16 and 27 and 28, respectively) were used to calculate the distance between the limbs and midbody imaginary line. First, we determined the midpoint of the shoulder and hip, landmarks 11 and 12 and 23 and 24, respectively, based on the midpoint formula:

$$(x_{mSh}, y_{mSh}) = \left( \frac{x_{11} + x_{12}}{2}, \frac{y_{11} + y_{12}}{2} \right) \tag{3}$$

$$(x_{mH}, y_{mH}) = \left( \frac{x_{23} + x_{24}}{2}, \frac{y_{23} + y_{24}}{2} \right) \tag{4}$$

where $(x_{mSh}, y_{mSh})$ represented the midpoint coordinate of the shoulder and $(x_{mH}, y_{mH})$ represented the midpoint coordinate of the hip. Then, to find the midbody imaginary line, we applied both midpoint (x, y) values to the line equation:

$$y = mx + c \tag{5}$$

substituting the y value of limb landmark coordinates (15 and 16 for the wrists, 27 and 28 for the ankles) to obtain the x value at the midbody imaginary line, followed by subtracting x of the landmarks to obtain the distance (Fig 1). All distance values were normalised to the average of the left and right body lengths (Euclidean distances of landmarks 11–23, 12–24). After obtaining the distance values from all the frames, we calculated the average distance of each of the four landmarks from the midline. Therefore, each video (120 s) had a single average number representing the distance from the midline of the four selected landmarks. Hence, the distance of landmark (Dlm) 15 represents the distance between landmark 15 and the midbody imaginary line, Dlm16 represents the distance between landmark 16 and the midbody imaginary line, and so on for Dlm27 and Dlm28.

## Procedure of pose estimation and finding body areas

In this study, our focus encompassed the entire body, including the upper and lower limb regions (Fig 2). Initially, we utilized the internal MediaPipe function in Python (version 3.9.12) for 2D pose estimation, resulting in 33 landmarks. Subsequently, we selected wrist and

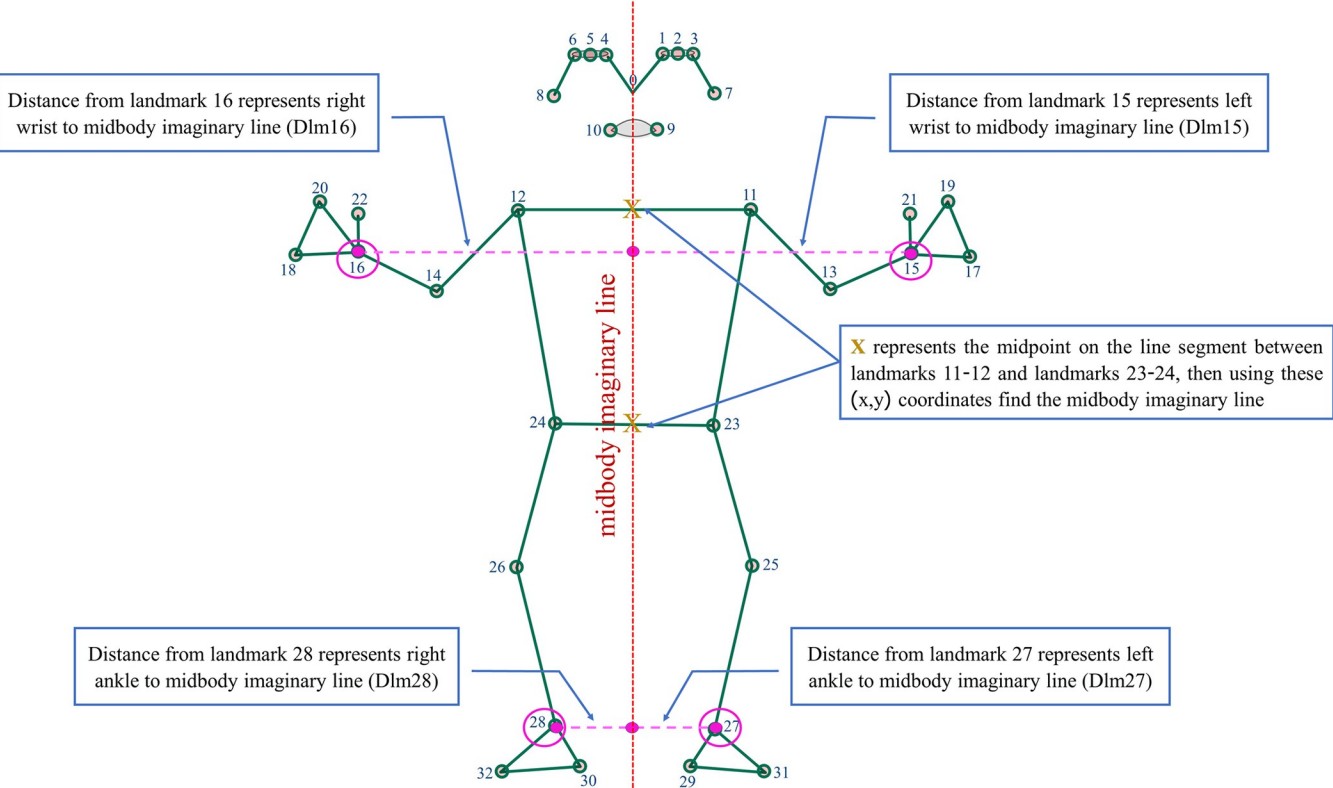

**Fig 1. Pose landmark [31] and calculated distance.** This figure shows the limbs' distance (dash line, —) from the landmarks 15, 16, 27, and 28 to the midbody imaginary line.

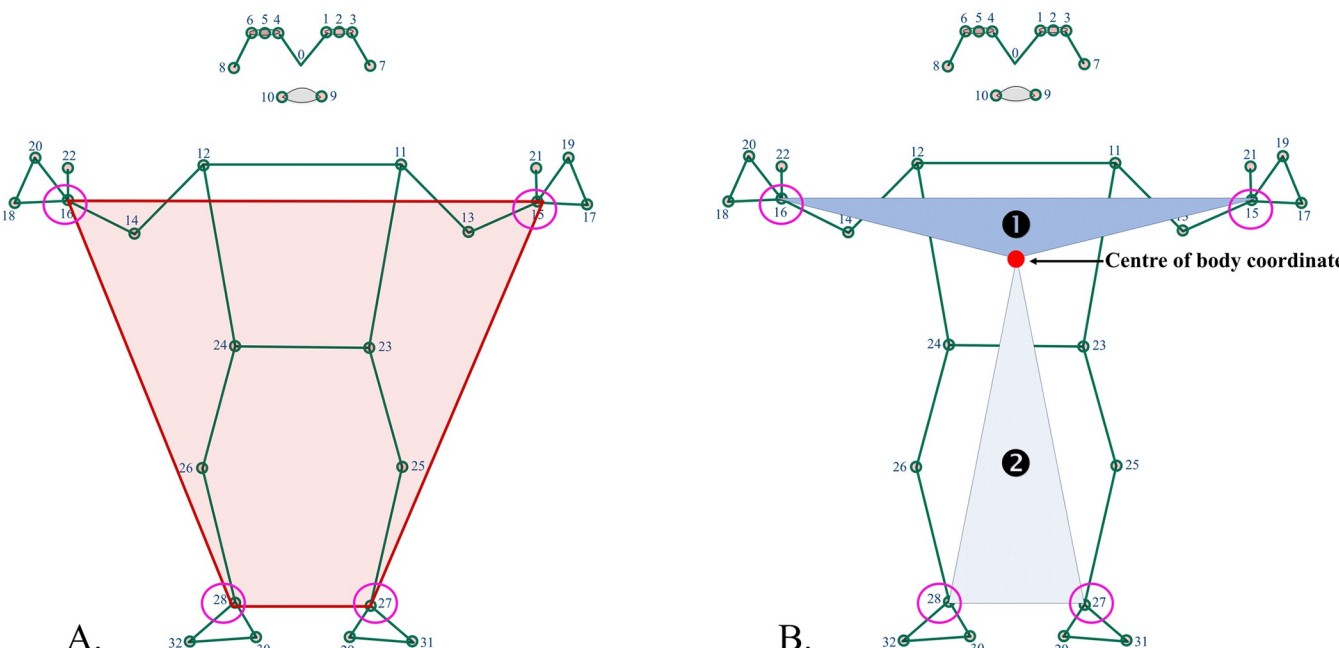

**Fig 2. Pose landmark [31] and calculated area.** This figure shows the calculated area using the limbs pose landmarks (15, 16, 27, and 28) and the centre of the body coordinate which are listed as: (**A**) presenting the whole area; (**B**) presenting ❶ = the upper part area, and ❷ = the lower part area.

ankle landmarks (15, 16, 27, and 28) to compute the quadrilateral area representing the whole area (Fig 2A). The centre of body coordinates were calculated using all landmark coordinates. Following this, we calculated the triangular area for the upper part using landmarks 15 and 16 and the centre of the body coordinate, as well as the lower part using landmarks 27 and 28 and the centre of the body coordinates (Fig 2B). All area values were normalized based on the average of the left and right body lengths. From each video, we computed the average area values for the whole area, upper part, and lower part.

## Data analysis

In this study, the values from the MTM observations were MTM occurrence percentage and MTM occurrence rate per min (total, lower limb, and upper limb), and pose estimations features were distance and area. Statistical analyses were performed using SPSS package for Windows version 26.0 (Armonk, NY: IBM Corp). Normality was assessed using the Kolmogorov–Smirnov Goodness of Fit test. The results were non-normally distributed except for whole area which showed normal distribution. The Spearman correlation test was used to calculate the correlation coefficient (r) between the average distance of the four landmarks to the midbody imaginary line and MTM observation values.

Then, we categorised the videos into two distinct groups: one featuring MTM and the other without MTM. Within these groups, we further classified the videos into three categories: those displaying Lower MTM; Upper MTM; and a combination of both, which we collectively labelled as 'Total MTM.' We employed an independent t-test to compare the whole area, whereas the Mann–Whitney U test was utilised to compare the upper- and lower-part areas. Statistical significance was defined as $p < 0.05$. The overall study methodology is depicted in Fig 3.

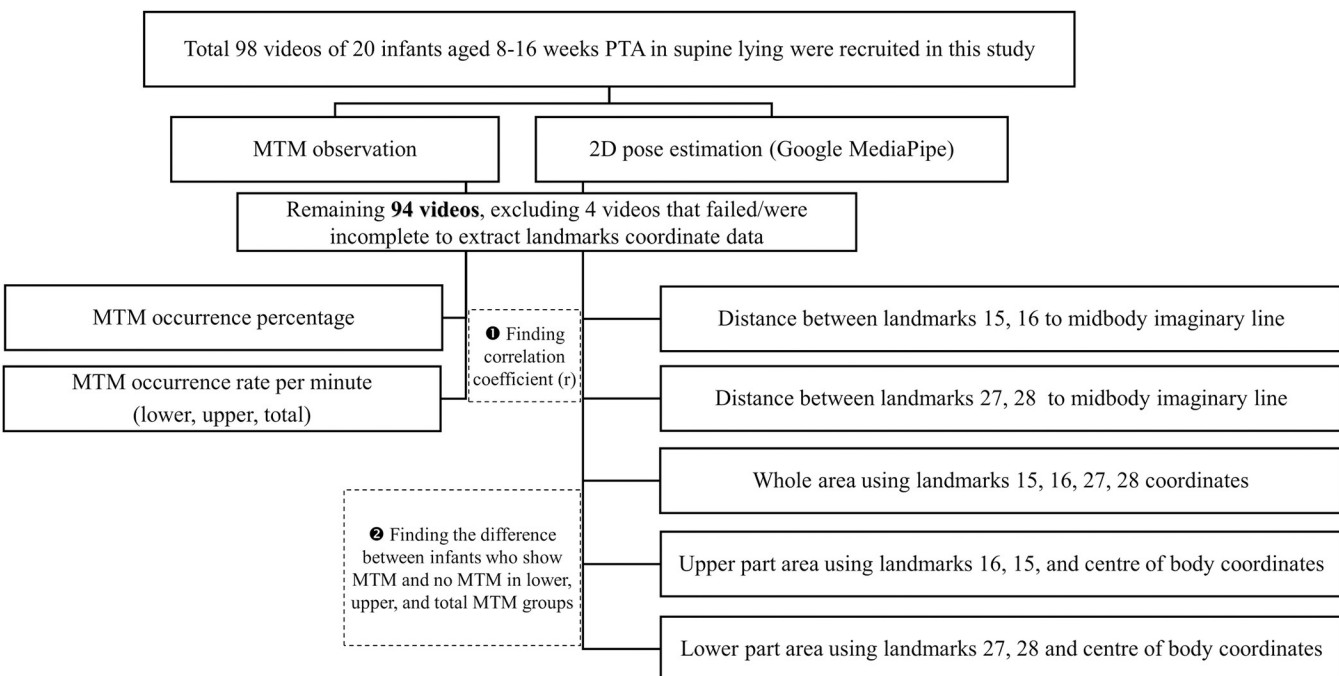

**Fig 3. Flow chart of the study methodology.** MTM = movement toward midline.

**Table 1. Anthropometric data at birth according to the 2006 WHO Child Growth Standards: 0–6 months (weight and length) and 0–13 weeks (head circumference) [32] (N = 20).**

|  | Gestational age (weeks + days) | Weight (g) (range of percentile) | Length (cm) (range of percentile) | Head circumference (cm) (range of percentile) |
|---|---|---|---|---|
| **Median** | 38 + 4 | 3150 (P15-P50) | 49 (P15-P50) | 34 (P50-P85) |
| **Min** | 37 + 0 | 2550 (P3-P15) | 47 (P3-P15) | 31 (<P3) |
| **Max** | 40 + 1 | 3900 (P85-P97) | 55 (>P97) | 36.5 (>P97) |

g = grams, cm = centimetres, P3 = 3rd percentile, P15 = 15th percentile, P50 = 50th percentile, P85 = 85th percentile, P97 = 97th percentile

## Results

A total of 98 videos (20 infants, aged 8–16 weeks PTA) were recruited for this study; four videos were excluded because of infants being out-of-frame or the videos being incomplete for landmark coordinate data extraction. Therefore, 94 infant videos were included in this study. Anthropometric data is summarised in Table 1. Gestational age (GA) was corrected for the term birth (40 weeks + 0 days). We reported birth weight, length, and head circumference with a range of percentile following 2006 WHO child growth standards [32]. The growth standard for weight and length (0–6 months), and head circumference (0–13 weeks), percentile lines are presented at the 3rd, 15th, 50th, 85th, and 97th percentile. Consequently, each infant's anthropometric data is reported within the corresponding percentile range. Moreover, the Apgar scores at min 1 and 5 are presented in S1 Table. The distribution of 10 MTM items are presented, with HT and FF contacts being the most frequently observed in the upper- and lower-limb MTM, respectively (Fig 4).

### Correlation between the calculated distance between limb landmarks and MTM observations

We investigated the relationship between the distance from upper limb landmarks to the mid-body imaginary line (Dlm15 for the left side and Dlm16 for the right side) and MTM

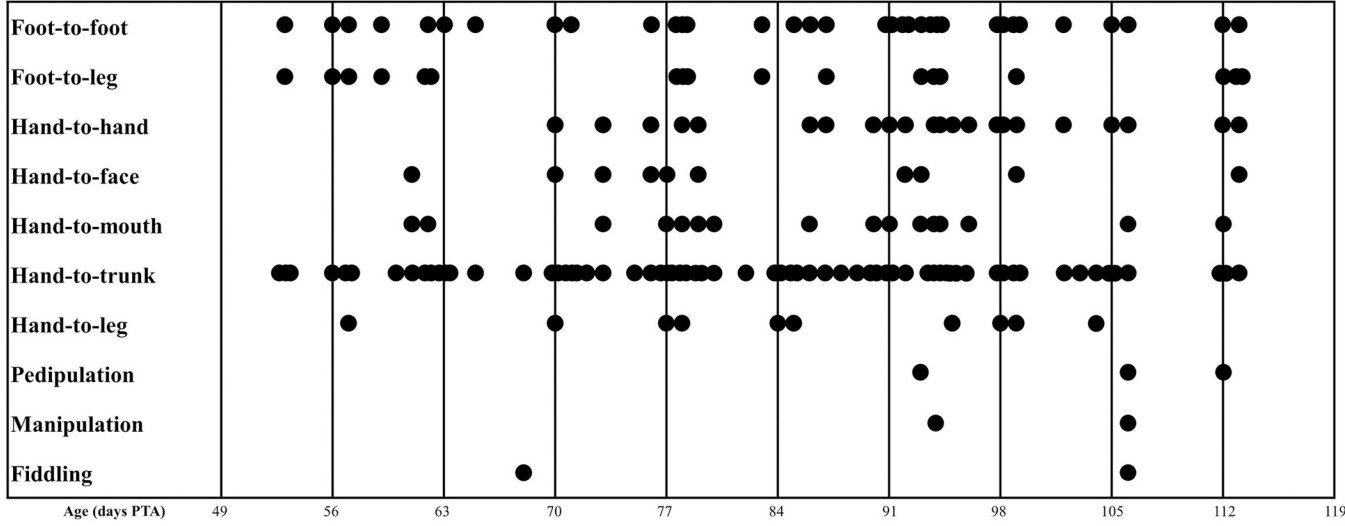

**Fig 4. The distribution of MTM frequency among infants who showed MTM on the recording day.** This figure shows the MTM occurrence data that was observed based on infant age in days post-term age (PTA). MTM = movement toward midline.

**Table 2. Correlation between distances from the four limb landmarks to the midbody imaginary line and MTM observation.**

| | Dlm15 | Dlm16 | Dlm27 | Dlm28 |
|---|---|---|---|---|
| Total MTM occurrence percentage | **0.315**\* (0.12, 0.4866) | **-0.354**\* (-0.519, -0.163) | -0.103 (-0.299, 0.1017) | **-0.331**\* (-0.5, -0.138) |
| Lower MTM occurrence rate per minute | 0.027 (-0.177, 0.2284) | -0.007 (-0.209, 0.1959) | -0.199 (-0.386, 0.0038) | **-0.354**\* (-0.519, -0.163) |
| Upper MTM occurrence rate per minute | **0.372**\* (0.1832, 0.5343) | **-0.393**\* (-0.552, -0.207) | -0.009 (-0.211, 0.194) | **-0.211**\* (-0.397, -0.009) |
| Total MTM occurrence rate per minute | **0.319**\* (0.1244, 0.4899) | **-0.360**\* (-0.524, -0.17) | -0.095 (-0.292, 0.1097) | **-0.334**\* (-0.503, -0.141) |

This table reported correlation coefficients (r) and their corresponding 95% confidence intervals in parentheses. MTM = movement toward midline, \* p-value < 0.01

Dlm15 represents the distance between landmark 15 and the midbody imaginary line

Dlm16 represents the distance between landmark 16 and the midbody imaginary line

Dlm27 represents the distance between landmark 27 and the midbody imaginary line

Dlm28 represents the distance between landmark 28 and the midbody imaginary line.

occurrence. Our findings showed that there were significant weak correlations between these distances and the total MTM occurrence percentage ($r_{Dlm15}$ = 0.315, $r_{Dlm16}$ = -0.354), as well as the occurrence rate per min for upper-limb MTM ($r_{Dlm15}$ = 0.372, $r_{Dlm16}$ = -0.393) and total MTM ($r_{Dlm15}$ = 0.319, $r_{Dlm16}$ = -0.360). Considering the distance for lower limb landmarks to the midbody imaginary line (Dlm27 for the left side and Dlm28 for the right side) and MTM occurrence, the distance from the right ankle landmark (Dlm28) showed significantly weak correlations to total MTM occurrence percentage ($r_{Dlm28}$ = -0.331), the occurrence rate per min for lower-limb MTM ($r_{Dlm28}$ = -0.354), and total MTM ($r_{Dlm28}$ = -0.334) (Table 2).

## Comparison of video groups that show MTM and no MTM

In the dataset, we found the following distribution of videos across three categories: Total MTM (82 videos showed MTM; 12 did not), Lower MTM (36 videos showed MTM; 58 did not), and Upper MTM (76 videos showed MTM; 18 did not). Significant differences were detected in both the whole area (p = 0.001) and lower part area (p = 0.003) within the Lower MTM group. However, no significant differences were found in area values observed for the Upper MTM and Total MTM groups (Fig 5).

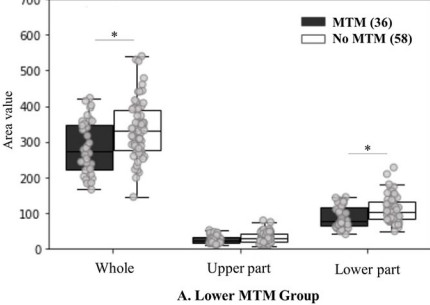 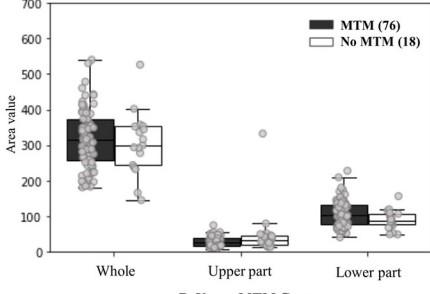 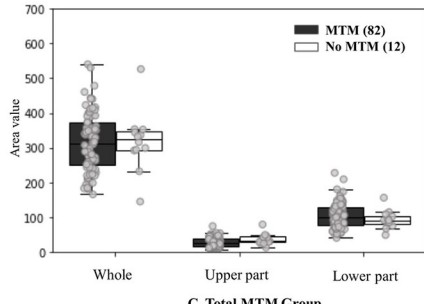

**Fig 5. The difference between groups of videos that showed MTM and those that did not.** (A) The Lower MTM group represents infants showing foot-to-foot contact, foot-to-leg contact, and pedipulation. The comparison of whole area and upper- and lower-part areas between the groups of videos that showed MTM and those that did not are presented. (B) The Upper MTM group represents infants showing hand-to-hand, hand-to-face, hand-to-mouth, hand-to-trunk, hand-to-leg, manipulation, and fiddling. The comparison of whole area and upper- and lower-part areas between the groups of videos that showed MTM and those that did not are presented. (C) The Total MTM group represents infants showing all 10 items: foot-to-foot contact, foot-to-leg contact, hand-to-hand, hand-to-face, hand-to-mouth, hand-to-trunk, hand-to-leg, pedipulation, manipulation, and fiddling. The comparison of whole area and upper- and lower-part areas between groups of videos that showed MTM and those that did not are presented. MTM = movement toward midline, \* p-value < 0.05. The sign of ′ represents the group of videos showing MTM and ≤ represents the group of videos not showing MTM.

## Discussion

In this study, we collected videos of healthy infants aged between 8- and 16-weeks PTA (recording by parents or guardians) and observed the occurrence of MTM within 10 MTM items (Lower MTM included FF, FL, and pedipulation, whereas Upper MTM encompassed HH, HF, HM, HT, HL, manipulation, and fiddling). We then utilised 2D pose estimation to extract landmark coordinates, enabling the calculation of distances and areas. There were three major findings of this study: 1) in the upper MTM, 'HT' was the most frequently observed motor behaviour, whereas in the lower MTM, 'FF' was the most prevalent; 2) we identified weak correlations between the calculated distance and MTM occurrence values; and 3) significant differences were observed in both the whole area and lower part area between videos showing MTM and those without MTM within the Lower MTM category.

MTM was considered the initial antigravity movement behaviour that would eventually develop into goal-directed movements, i.e. reaching and grasping, feeding, and walking. In this study, the FF and HT movement patterns were most frequently observed for the lower and upper limbs, respectively. In typical infants, lower limb movement normally begins with leg lifting and then progresses to foot contact and exploration with various patterns of leg movements [33, 34]. In this study, three out of ten MTM items exhibited the development of lower limb movements over time. Consequently, the frequency of FF occurrences was notably high in this age group. This observation is consistent with a previous study [14], which demonstrated a similar increase in the frequencies of FF by age. FF movement presented as an early antigravity movement; the absence of FF potentially identifies neurodevelopmental problems during early infancy. Ferrari et al. [35] studied the use of FF movements for supporting FMs by assessing both FMs and FF occurrence in infants with hypoxic-ischemic encephalopathy, describing the sensitivity and specificity for CP prediction individually and when pooling the two factors together. The result showed the sensitivity and specificity of both patterns to predict CP was 100% and 73%, respectively. This suggests that FF could serve as a supportive tool, emphasizing the need for further exploration of various FF movements. When considering FF subtypes, such as plant-to-plant contact, plant-to-tibial margin of the contralateral foot, plant-to-dorsal part of the contralateral foot, feet crossing, and pedipulation, more specific information about infant development can be obtained [35]. Meanwhile, the FL components exhibit higher development, involving lifting the leg and crossing over to touch the opposite side. The repetition of this movement pattern could lead to the rolling component [36]. HT is a fundamental motor behaviour observed in infants when they exhibit fewer uncontrolled movements (i.e. swipes and oscillations) or it could be from ATNP. Among the seven upper limb MTM items, it is challenging to deduce that HT is an antigravity movement of the entire arm; certain components involve lifting the forearm from the floor and placing it against the body. However, infants brought their arms and hands to contact their body to initiate self-exploration. This motor behaviour can enhance somatosensory input, skill development, and subsequent behaviours [37, 38].

Considering the correlation between the distance calculated from upper arm landmarks and MTM, occurrence values revealed significant positive correlations with left wrist landmarks and negative correlations with right wrist landmarks. These associations may be attributed to the infant's posture, where one hand contacts the body while the other remains apart. After birth, infants experience a different environment as they are generally used to being in a flexion position with an inability to move against gravity. When they are lying on their back, they may show a variety of motor and posture patterns [9]; head turn aside, arm sweeping, or leg kicking unintentionally. In this study, the different directions of upper limb landmarks may represent the asymmetry which is commonly found in early infant development. In

addition, the presence of ATNP may influence this outcome, as ATNP occurs when the infant rotates their head to the side while extending one arm and leg and flexing the other arm and leg on the opposite side [14, 39]. As infant postural control improves with age, various MTM items, such as HH, begin to emerge and develop symmetrical movement patterns [40].

In this study, we calculated area using wrist and ankle landmarks. According to the hypothesis, the presence of MTM may decrease these areas. In the lower-limb MTM group, significant differences were observed in the whole and lower area between videos that showed MTM and no MTM. The lower limb MTM category encompassed FF, FL, and pedipulation movements. These specific movements involve positioning the ankle and foot closer to each other, resulting in a smaller area compared with having the legs apart. Infants require strength, skill, and experience to bring their feet together while lifting their legs [41, 42]. In the upper-limb MTM group, there were no significant difference in all areas between videos showing MTM and no MTM. However, when considering the upper part area, the group displaying MTM exhibits a smaller area compared with those not showing MTM. This category encompasses seven MTM items that can be divided into two groups: one hand and both hands contacting body parts. These movements led to varied positions, including both wrist landmarks being close together and one landmark apart from another; the latter configuration resulted in a larger area. Further investigations could involve grouping arm positions and comparing corresponding areas to gain a more comprehensive understanding of the subject matter.

Automated pose estimation has been employed in many studies assessing early infant movement development. For instance, Marchi et al [43] reported high accuracy using Open-Pose skeletal videos compared with visual assessments for GMA. Turner et al [44] applied a combination of deep learning and 2D pose estimation to classify children's dexterity and position when interacting with a toy. In this study, we employed the open-source Google Media-Pipe 2D pose estimation to identify features related to MTM in the supine position. Analysing distance and area revealed certain associations between computed values and MTM observation. Regarding the antigravity characteristic of MTM movement, a more in-depth investigation is warranted to uncover additional insights. Further complexity analysis, such as the development of 2D landmarks into 3D pose estimation [23], may be necessary to enhance our understanding. However, assessing MTM movement in infants aged between 3 and 4 months could be practical in clinical settings [14]. The implementation of pose estimation may be useful for infants in remote areas or facing challenges with transportation to hospitals or rehabilitation centres.

This study had some limitations. First, video recording was done by a parent or guardian; hence, it was common to have some interferences that could not be avoided due to their presence when approaching infants of this age. Infants might react and need a response from the person who was taking a video. Some of them stared at the mobile phone and did not show any movement. Second, despite efforts to maintain a perpendicular position for handheld smartphones during recording, slight movements or changes in camera angles owing to the camera-holding hand were possible. However, since the features calculated in this study are angular information obtained by connecting markers with lines and distance information corrected by body length, we believe that their influence was minimized. Hand-holding a camera may have been a convenient and realistic method for parents or caregivers. However, if they had to set up a camera tripod or control the camera similar to a laboratory setting, this may have become burdensome. Improving the capacity to adapt to factors such as camera angle and occlusion is essential for optimizing the functionality of a lightweight, parent-controlled system in identifying neuromotor delays in infants. This enhanced ability ensures more robust performance, enabling accurate assessments even in challenging conditions. Third, the sample size appears to be limited, and as a result, the presented findings may not accurately represent

all infants. Further study is required with a larger number of infants and an extended duration of follow-up to thoroughly assess the effectiveness of the 2D pose estimation method in extracting values related to MTM.

## Conclusions

Assessing movement in infants at an early age plays an important role in detecting motor disorders. The emergence of MTM during spontaneous movement may support intentional movement development. The novel approach of markerless movement analysis has the potential to enhance the developmental follow-up system. In this study, infant video recordings were captured by parents or guardians. Open source 2D pose estimation features (coordinates, distance, and area) were utilised to provide support in identifying the presence of MTM in infants. We observed a significant correlation between MTM observation values and distances calculated from limb landmarks to the midbody imaginary line. Notably, both the whole area and the lower part area exhibited significantly lower values in the videos showing MTM, especially in the Lower MTM group. This method offers advantages for natural infant movement with non-invasive techniques, lightweight computational analysis, and regular follow-up, particularly in isolated areas or difficult situations. However, the development of 2D landmarks into 3D pose estimation is challenging due to the complexity of MTM movements characterized by antigravity patterns. In future studies, a larger sample size and inclusion of high-risk infants are essential to assess MTM patterns and differentiate between pathological and typical MTM movements.

## Supporting information

**S1 Table. Infant demographic data (N = 20).** This table included gestational age (GA) and anthropometrics at birth. Percentiles are presented according to the 2006 WHO Child Growth Standards: 0–6 months (weight and length) and 0–13 weeks (head circumference) [32]. The Apgar scores at 1 and 5 minutes, along with the age at the time of video collection, are provided for each infant.
(DOCX)

## Acknowledgments

We extend our heartfelt gratitude to Dr.Nuttawat Rungsirisilp and Dr.Kornkanok Tripanpitak for their invaluable support during the pose estimation and feature extraction processes.

## Author Contributions

**Conceptualization:** Nisasri Sermpon, Hirotaka Gima.

**Formal analysis:** Nisasri Sermpon, Hirotaka Gima.

**Funding acquisition:** Nisasri Sermpon, Hirotaka Gima.

**Investigation:** Nisasri Sermpon, Hirotaka Gima.

**Methodology:** Nisasri Sermpon.

**Project administration:** Nisasri Sermpon, Hirotaka Gima.

**Resources:** Nisasri Sermpon, Hirotaka Gima.

**Supervision:** Hirotaka Gima.

**Visualization:** Nisasri Sermpon, Hirotaka Gima.

**Writing – original draft:** Nisasri Sermpon.

**Writing – review & editing:** Nisasri Sermpon, Hirotaka Gima.

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
