## [Decision Letter · Decision Letter 0]

13 Dec 2023

PONE-D-23-29999Correlation between pose estimation features regarding movements towards the midline in early infancyPLOS ONE

Dear Dr. Gima,

Thank you for submitting your manuscript to PLOS ONE. After careful consideration, we feel that it has merit but does not fully meet PLOS ONE’s publication criteria as it currently stands. Therefore, we invite you to submit a revised version of the manuscript that addresses the points raised during the review process.

We look forward to receiving your revised manuscript.

Kind regards,

Prince Jain

Academic Editor

PLOS ONE

Journal Requirements:

Additional Editor Comments:

Your work has been critically evaluated by the reviewers, and it currently lacks the required quality for publication. It is strongly advised to carefully consider and incorporate the reviewers' recommendations. If you believe you can adequately address the reviewers' comments, which are outlined below, I invite you to revise your manuscript thoroughly and resubmit it for reconsideration.

Reviewers' comments:

Reviewer's Responses to Questions

**Comments to the Author**

1. Is the manuscript technically sound, and do the data support the conclusions?

Reviewer #1: Yes

Reviewer #2: Partly

Reviewer #3: No

2. Has the statistical analysis been performed appropriately and rigorously? 

Reviewer #1: Yes

Reviewer #2: No

Reviewer #3: No

3. Have the authors made all data underlying the findings in their manuscript fully available?

Reviewer #1: Yes

Reviewer #2: No

Reviewer #3: Yes

4. Is the manuscript presented in an intelligible fashion and written in standard English?

Reviewer #1: No

Reviewer #2: Yes

Reviewer #3: Yes

5. Review Comments to the Author

Reviewer #1: See attachments (2)

Reviewer #2: The study uses of 2D pose estimation for analyzing infant movements. However, significant revisions are needed to address the concerns raised, particularly regarding methodology, data analysis, and the discussion of results. Main concerns are:

1. The Introduction section should more effectively set the stage for the research, outlining the current state of knowledge, identifying gaps, and explicitly stating the research question or hypothesis.

2. The description of the methodology, particularly the video recording and data collection procedures, lacks comprehensive details. For instance, there is no mention of the camera specifications used for recording, which could significantly impact the pose estimation accuracy. Additionally, information on the calibration process for the 2D pose estimation tool and the handling of potential errors in pose estimation is missing.

3. The sample size appears limited (20 infants), raising concerns about the generalizability of the findings. Also, the criteria for selecting the infants and the videos used for analysis are not adequately detailed.

4. The discussion section requires expansion. It should not only interpret the results but also compare them with existing literature, discussing how this study contributes to the current understanding of infant movement analysis.

5. The study's limitations are not thoroughly discussed.

6. Some references are outdated or not directly relevant to the study's focus. Updating these with more recent and pertinent studies would strengthen the background and support the research.

Reviewer #3: Remarks:

The study seems an interesting novel approach which is appreciable, yet there are major concerns regarding the scientific soundness ,validity, reliability and external journalisability of the study.

The authors seem to be representing an over simplified and speculative version of the findings, which may lead to huge erreneous impressions to the scientific community.

There is no mention of the study method, settings, sample size estimation and validity of the tools used in the scientific methodology of the current study.

In fact the previous literature which have been referred to as the hypothesis generating resource,e.g ref 11 Ferrari et al, also have major limitations in terms of an appropriate size of sample used in their study to be used for anaysis of a statistically and clinically relevant study.Further the previous authors have clearly mentioned about the unreliability of the time line used for analysis in their study and have recommended strongly for the recordings at later time period of development as a future recommendation, which yet has been ignore in the current study method.

The population used in the previous literature are the children suffering from hypoxic ischemic encephalopathy on which the findings of investigation are based upon, the current study method has conveniently ignored a very important confounder that is the neurological health status of the infant in this investigation including that could have talked of a possibly entire different scenario.

In my humble opinion the study does not seen fit for publication in the current format.

In fact at any other point of time if the authors are interested to submit this to any other journal it must be referred to as preliminary observations only or the the scientific soundness of the study methodology needs to be upgraded and then revised findings should be published at a later time.

6. PLOS authors have the option to publish the peer review history of their article (what does this mean?). If published, this will include your full peer review and any attached files.

Reviewer #1: **Yes: **Manon Maitland Schladen

Reviewer #2: No

Reviewer #3: **Yes: **Vishakha Grover

---

## [Author Response · Author response to Decision Letter 0]

26 Jan 2024

Thank you for your consideration. Please see the attached document 'Response to Reviews'.

---

## [Decision Letter · Decision Letter 1]

16 Feb 2024

Correlation between pose estimation features regarding movements towards the midline in early infancy

PONE-D-23-29999R1

Dear Dr. Gima,

We’re pleased to inform you that your manuscript has been judged scientifically suitable for publication and will be formally accepted for publication once it meets all outstanding technical requirements.

Kind regards,

Prince Jain

Academic Editor

PLOS ONE

Additional Editor Comments (optional):

Reviewers' comments:

Reviewer's Responses to Questions

**Comments to the Author**

1. If the authors have adequately addressed your comments raised in a previous round of review and you feel that this manuscript is now acceptable for publication, you may indicate that here to bypass the “Comments to the Author” section, enter your conflict of interest statement in the “Confidential to Editor” section, and submit your "Accept" recommendation.

Reviewer #1: All comments have been addressed

Reviewer #3: (No Response)

2. Is the manuscript technically sound, and do the data support the conclusions?

Reviewer #1: Yes

Reviewer #3: Partly

3. Has the statistical analysis been performed appropriately and rigorously? 

Reviewer #1: Yes

Reviewer #3: I Don't Know

4. Have the authors made all data underlying the findings in their manuscript fully available?

Reviewer #1: Yes

Reviewer #3: Yes

5. Is the manuscript presented in an intelligible fashion and written in standard English?

Reviewer #1: Yes

Reviewer #3: Yes

6. Review Comments to the Author

Reviewer #1: Thank you for your very thorough response to comments, mine and those of the other reviewers. It has been a pleasure to read your work. I look forward to being able to share it, once published, with my colleagues who are also involved in infant movement analysis in in the home.

Reviewer #3: Dear Authors

The manuscript is much improved than earlier, can be considered. However, yet the limitations as remarked earlier,need to be highlighted in the end of conclusion.

7. PLOS authors have the option to publish the peer review history of their article (what does this mean?). If published, this will include your full peer review and any attached files.

Reviewer #1: **Yes: **Manon Maitland Schladen

Reviewer #3: No

---

## [Editor Report · Acceptance letter]

19 Feb 2024

PONE-D-23-29999R1 

PLOS ONE

Dear Dr. Gima, 

I'm pleased to inform you that your manuscript has been deemed suitable for publication in PLOS ONE. Congratulations! Your manuscript is now being handed over to our production team.

Kind regards, 

on behalf of

Dr. Prince Jain 

Academic Editor

PLOS ONE